# Size Matters: Rethinking Hertz Model Interpretation for Cell Mechanics Using AFM

**DOI:** 10.3390/ijms25137186

**Published:** 2024-06-29

**Authors:** Katarína Mendová, Martin Otáhal, Mitja Drab, Matej Daniel

**Affiliations:** 1Department of Mechanics, Biomechanics and Mechatronics, Faculty of Mechanical Engineering, Czech Technical University in Prague, Technická 4, 16000 Prague, Czech Republic; katarina.mendova@fs.cvut.cz; 2Department of Natural Sciences, Faculty of Biomedical Engineering, Czech Technical University in Prague, Náměstí Sítná 3105, 27201 Kladno, Czech Republic; martin.otahal@fbmi.cvut.cz; 3Laboratory of Biophysics, Faculty of Electrical Engineering, University of Ljubljana, Trzaska 25, 1000 Ljubljana, Slovenia; mitja.drab@fe.uni-lj.si

**Keywords:** atomic force microscopy (AFM), cell mechanics, cell stiffness, Hertz contact model

## Abstract

Cell mechanics are a biophysical indicator of cell state, such as cancer metastasis, leukocyte activation, and cell cycle progression. Atomic force microscopy (AFM) is a widely used technique to measure cell mechanics, where the Young modulus of a cell is usually derived from the Hertz contact model. However, the Hertz model assumes that the cell is an elastic, isotropic, and homogeneous material and that the indentation is small compared to the cell size. These assumptions neglect the effects of the cytoskeleton, cell size and shape, and cell environment on cell deformation. In this study, we investigated the influence of cell size on the estimated Young’s modulus using liposomes as cell models. Liposomes were prepared with different sizes and filled with phosphate buffered saline (PBS) or hyaluronic acid (HA) to mimic the cytoplasm. AFM was used to obtain the force indentation curves and fit them to the Hertz model. We found that the larger the liposome, the lower the estimated Young’s modulus for both PBS-filled and HA-filled liposomes. This suggests that the Young modulus obtained from the Hertz model is not only a property of the cell material but also depends on the cell dimensions. Therefore, when comparing or interpreting cell mechanics using the Hertz model, it is essential to account for cell size.

## 1. Introduction

In the past 30 years, atomic force microscopy (AFM) has revolutionized the way we probe cell mechanics [1]. Originally developed to provide topographic images of solid surfaces [2], it is now routinely applied to measure the mechanical properties of individual cells [3]. AFM quantifies cell mechanics by applying a subtle but controlled force or displacement to a cell through a tiny tip and measuring the response with high precision [4].

Unlike other methods such as fluorescent microscopy, cryo-electron tomography, or three-dimensional electron microscopy, AFM allows cells to be studied without staining, labeling, or fixation, thus under physiological conditions [3]. Cell mechanics are observed to indicate cell biological functions such as adhesion, migration, or differentiation [1,5,6]. The mechanical properties of cells are also related to pathology, particularly metastatic cancer [7], cardiovascular disease [8], or infection by microbes or viruses [9].

During the development of the AFM technique, considerable effort was put into developing fast, accurate, and gentle microscopes. The high-speed imaging of living cells permits the study of drugs in surface cell structures [1] or the mechanical mapping of subcellular and subnuclear structures in real time [10]. Although progress in instrumentation and method has been considerable, little has changed in the processing and analysis of AFM results.

In mechanical analysis, the response of the material to external stimuli, such as applied fields or forces, is expressed by constitutive parameters [11]. These are quantities that are specific to each material. In theory, the constitutive parameters should be independent of the measurement instrument, sample, method, or model. In a first approximation, the cell could be considered a homogeneous isotropic elastic material and hence characterized by two parameters, e.g., Young’s modulus *E*, and Poisson ratio ν [12]. Young’s modulus describes how a cell material resists deformation when uniaxial stress is applied, while Poisson’s ratio is a measure of how much the cell deforms in the lateral direction when compressed in the axial direction. Since the Poisson ratio of most soft biological tissues is very close to 0.5 [13] and the error in Young’s modulus due to the unknown Poisson ratio is less than 10% [14], the cell mechanics are usually characterized only by the Young modulus [12].

The AFM provides a non-linear force/displacement curve even for elastic engineering materials that are conditioned by non-linear contact mechanics between the small tip and large sample. To extract the elastic modulus from AFM experiments, a model of contact mechanics is usually employed. The most common models are the Hertz and the Sneddon models for spherical and conical indenters, respectively [15,16]. The general description of indentation curve analysis was provided later by Pharr, Oliver, and Brotzen, 1992 [17].

It has been shown that both the Hertz model and Oliver–Pharr model provide identical results for purely elastic samples and for spherical indenters [18]. For the Hertz model, the dependence between the indentation force *F* and indentation depth *h* for a stiff spherical tip of radius *R* is expressed as [19]:$$F=\frac{4}{3}E^{*}\sqrt{R}h^{3/2}$$
where $E^{*}$ is the reduced Young modulus:$$E^{*}=\frac{E}{1-\nu^{2}}$$

The derivation of the Hertz model along with its assumptions are outlined in Appendix A. The basic assumptions of the Hertz model are that the strains are small and within the elastic limit, which means that the cell material behaves linearly and recovers its original shape after the contact; the surfaces of the cell and indenter are continuous and nonconforming, which means that the contact area is much smaller than the characteristic dimensions of the cell and the tip; the cell surface is frictionless, which means that there is no tangential force or shear stress between the surfaces of the AFM tip and cell; and the AFM tip is absolutely stiff with the shape of an exact sphere, which means that deformations of the tip and substrate are negligible compared to cells. Deviation from these assumptions in live cells given by the geometry, composition, and material heterogeneity of the cells brings additional variability into the estimated Young’s modulus [20]. It is further assumed that if the cell is much larger than the depth of indentation, the stresses induced by loading vanish far from the indentation point. As shown in Appendix A, this assumption considers the cell to be an elastic half-space, meaning the cell volume is infinitely large compared to the area of contact. The elastic half-space assumption is a cornerstone for both the Hertz and Oliver–Pharr models [21].

Although theoretical and experimental studies based on elastic shell theory [22] and liquid drop theory [23] suggest that cell stiffness is considerably affected by its dimensions, this factor is neglected in Hertz theory. We hypothesize that larger cells will have lower stiffness and therefore a smaller Young’s modulus, and vice versa. To minimize the effect of mechanical and shape variability between individual cells, we will use liposomes as cell models and evaluate Young’s modulus by fitting the standard Hertz model to the experimental AFM force/deflection curves as outlined in Figure 1.

## 2. Results

The Hertz model provides a good fit to the AFM data as evidenced by the high correlation coefficient between the measured and predicted values (Pearson correlation coefficient mean 0.998 for all measurements, range 0.9881–0.999). Representative loading curves are presented in Figure 2. Young’s modulus in HA-filled liposomes (mean 1.11 kPa, range 0.30–1.85 kPa) is significantly higher than in PBS-filled liposomes (mean 0.37 kPa, range 0.62–1.28 kPa) (Wilcoxon rank sum test, *W* = 423, p< 0.001). The higher stiffness in HA-filled liposomes corresponds to steeper force/deflection curves (Figure 2). The results showed a high degree of agreement between repeated measurements as indicated by the low variation between the measured curves and in the estimated Young modulus (Figure 2 and Figure 3).

Linear regression was used to test whether the liposome size significantly predicts Young’s modulus. For both liposomes filled with PBS and HA, the effect of the liposome diameter *d* is statistically significant and negative (β = −23.44, 95% CI [−28.33, −18.56], p< 0.001 for the liposome filled with PBS and β = −36.53, 95% CI [−45.58, −27.48], p< 0.001 for liposomes filled with HA). The effect of the diameter of the liposome on Young’s modulus is significantly higher for HA-filled liposomes (ANOVA p=0.008).

## 3. Discussion

One of the primary assumptions of the Hertz model for analyzing contact mechanics is that the cell is elastic, isotropic, and homogeneous, and that the indentation is small compared to the size of the cell [19]. In this study, we evaluated the effect of cell size on the estimated Young’s modulus using liposomes as cell models, and adopting methods proposed for cell mechanics measurements [24]. We demonstrated a significant dependence between the size of the measured liposome and its stiffness (Figure 2 and Figure 3).

Our findings are consistent with previous studies by Delorme et al. (2006) [25], who observed higher stiffness in smaller liposomes. The observed size effect of liposomes aligns with the shell theory of cell deformation [26,27]. Real-time deformability cytometry also indicates greater deformation for larger cells of the same phenotype [28] as observed in our study.

The scattering of data in Figure 3 suggests the influence of additional factors on the measured mechanical response. One such factor that warrants further evaluation is liposome adhesion. Theoretical [29] and experimental studies [30] have shown that extensive cell adhesion increases cell membrane tension and stiffness. AFM measurements also indicate that the stiffness of adherent epithelial cells increases with the projected area of apical cells [31]. Hence, liposomes with significant adhesion to the surface were excluded from our analysis. Overbeck et al. (2017) demonstrated that osmotic pressure can also affect cell response, with higher osmolarity contributing to decreased cell stiffness [23].

For spherical probes, it has been reported that the sample can be considered an elastic half-space if the indenter’s radius is at least ten times smaller than each horizontal dimension of the sample [15]. The Oliver–Pharr indentation analysis further suggests that the Hertz model is applicable for spherical indenters when the h/R ratio is less than 10 [18]. Modifications to the Hertz model have been proposed to introduce correction factors for substantial deformations [18].

The presented results were obtained using liposomes as cell models, which reduces variability in input parameters by employing experimental samples with controlled composition and geometry. While liposomes mimic basic cell structures, they may not fully capture the active behavior of living cells. Cells are heterogeneous structures with intricate internal organization. For instance, the anisotropy of the cytoskeleton induces non-axisymmetric deformations [32], and the stiffness of subcellular structures influences local mechanical responses [10]. The prolonged or repeated indentation of single cells can lead to cytoskeletal remodeling [33], which further impacts cell stiffness. When measuring live cells, active responses due to cytoskeletal remodeling should also be considered [34]. The experimental model could be enhanced by incorporating self-assembled actin shells [35].

To create a more realistic artificial cell model, we tested HA-filled liposomes. The high molecular weight and semi-flexible chain of HA impart viscous and elastic properties [36], akin to those observed in the cytoplasm [37]. The viscosity of the HA solution used (100 Pa s) [38] corresponds to the viscosity of cell cytoplasm [39]. The addition of HA, mimicking cytoplasm, increases the effect of cell size on the estimated Young’s modulus. Our results underscore the importance of considering the internal environment in modeling cell mechanics. Further research is needed to quantify the relationship between cell size and stiffness in confluent and highly adherent living cells with complex internal organization.

## 4. Materials and Methods

### 4.1. Liposomes Preparation

Liposomes are prepared using a two-stage microfluidic device that produces filled liposomes using the double emulsion drop method. The custom-made microfluidic device is manufactured using PolyJet technology (Polyjet J750, Stratasys, Eden Prairie, MN, USA) from VeroClear-RGD810. The device consists of an inner aqueous phase channel, two lipid-carrying organic phase channels, an intermediate channel, two outer aqueous phase channels, and a downstream channel [40]. Using volume-controlled flow pumps, the inner aqueous stream of HA (molecular weight 2000–2200 kDa, concentration 5:1, Contipro, DolníDobrouč, Czechia) is dissolved in PBS (200 mL) for filled liposomes or PBS (10 mL) for PBS-filled liposomes. The surrounding lipid-carrying streams (DPPC in isopropyl alcohol, Sigma-Aldrich, Burlington, MA, US) are hydrodynamically focused, and a single emulsion droplet is formed by shearing the inner phase. Subsequently, a double emulsion is formed by two external streams of aqueous PBS (10 mL). As the aim of the study is to produce liposomes of various sizes, the diameter of the inner channel is 0.5 mm, allowing the formation of multidispersed liposomes. Liposome formation is driven by shear flow at the junctions by setting the individual flow rate ratio at 5, 10 and 15 mL/h for phospholipids, inner fluid and outer fluid, respectively. Liposome formation and flow focusing are inspected in situ using a phase microscope (NIB-100 Inverted Microscope with Canon SCR Camera, Canon, Tokyo, Japan).

### 4.2. Liposomes Fixation

The binding required to measure the mechanical properties of liposomes by AFM was achieved using an avidin–biotin complex. Biotin–DOPE and DPPC lipids were used at a concentration ratio of 1:1000. The biotinylated surface was incubated with avidin (0.30 mg/mL) and washed three times with PBS buffer. Subsequently, 1 mL of the liposomal formulation and approximately 1 mL of PBS buffer were applied to a Petri dish and incubated for 5 min at room temperature [41,42].

### 4.3. AFM Measurements

Mechanical testing of the liposomes was performed using the NanoWizard^®^ 3 NanoOptics AFM System (JPK, DE). A colloidal probe with a diameter of 5.2 μm and a spring constant of 0.0307 N/m (APPnano, Mountain View, CA, USA) was employed. The cantilever was calibrated according to the manufacturer’s standard procedure. First, the sensitivity of the cantilever was determined via indentation measurements on glass, followed by determining the stiffness using thermal noise analysis. Force spectroscopy of the liposomes was performed with a z length of 15 μm, a relative set point of 20 nN, and a loading rate of 3.75 μm/s.

The following inclusion criteria were applied: the isolated spherical shape of the liposome without collapse [43] or extensive adhesion to the surface [44], and at least two successful measurements per liposome. Force/deformation curves were measured at the center of each liposome.

### 4.4. Data Processing

After subtracting the deflection of the cantilever, the force/displacement curves were fitted using the Hertz model for a hemispherical cantilever tip with and cell Poisson ratio of 0.5, according to the method described by Thomas et al. (2013) [24]. Data indicating strong attachment of liposomes to the surface (small height compared to diameter), liposome burst (rapid change in force), or extensive noise (AFM tip fouling) were excluded from the analysis. The final analysis included 162 measurements (116 and 46 in PBS- and HA-filled liposomes, respectively) across 46 liposomes (31 and 15 in PBS- and HA-filled liposomes, respectively).

### 4.5. Statistical Analyses

Statistical analyses were performed using R software (version 4.1.2, R Core Team, 2021). A *p* value less than 0.05 was considered statistically significant. The normality of the data was tested using the Shapiro–Wilk test. Differences in Young’s modulus between different liposome types or treatments were analyzed using one-way analysis of variance (ANOVA), followed by Tukey’s post hoc test for normally distributed data, and by the Wilcoxon rank sum test otherwise. The correlation between Young’s modulus and cell size was assessed using linear regression (package lme4), considering repeated measurements [45]. In total, 95% confidence intervals (CI) and *p* values were calculated using a Wald *t* distribution approximation.

## 5. Conclusions

Atomic force microscopy (AFM) provides estimations of cellular stiffness. To attain statistical robustness in the analysis of highly heterogeneous cellular populations, a comprehensive examination of numerous individual cells is essential [46]. Moreover, the versatility of AFM encompasses the determination of cellular states through assessments at the single-cell level [47]. Enhancing the reliability of single-cell mechanics techniques necessitates the identification of potential variability sources among cells. Variability arises not only from physiological disparities but also from the methodologies employed in testing and subsequent data processing. Our empirical data suggest that cellular size, an aspect not accounted for in the foundational assumptions of the Hertz model, considerably influences the measured stiffness and the inferred Young’s modulus. Consequently, for comparative analyses, it is advisable to select cells with commensurate dimensions. Further research is required to develop a correction factor tailored to the unique characteristics of living cells with complex internal structures.

## Figures and Tables

**Figure 1 ijms-25-07186-f001:**
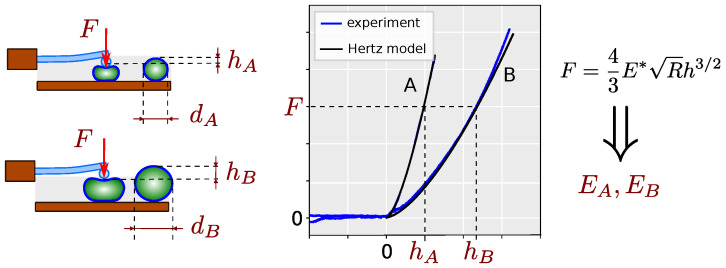
Overview of study: AFM measurement of liposomes of various sizes: A small diameter, B large diameter; curves fitting with Hertz contact model; estimation of Young’s modulus of elasticity as fitting parameter.

**Figure 2 ijms-25-07186-f002:**
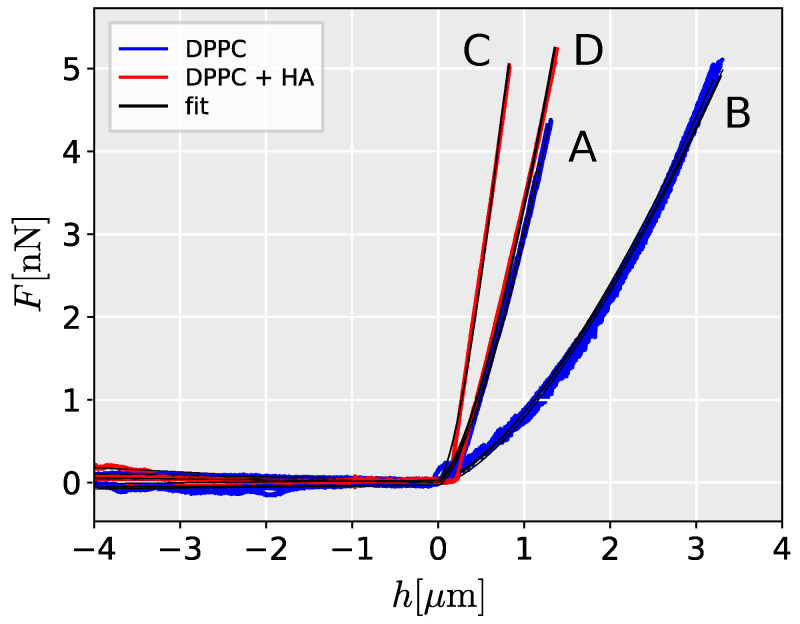
Measured indentation curve for DPPC liposomes in PBS filled with (DPPC) PBS and (DPPC + HA) HA solution. The fit of indentation by the Hertz contact model for the hemispherical AFM tip is shown. Refer to Figure 3 for details on individual liposomes properties (capital letters).

**Figure 3 ijms-25-07186-f003:**
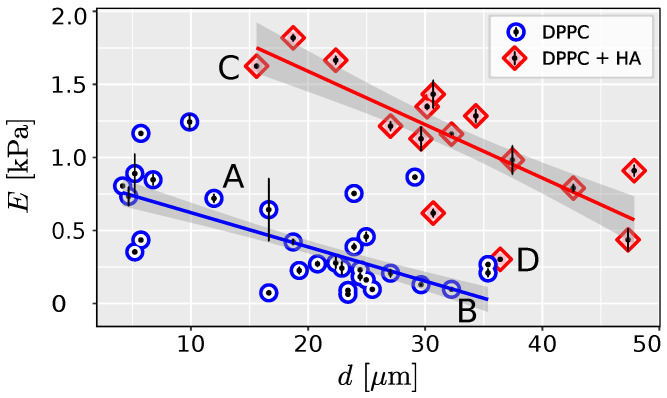
Linear regression plot with 95% confidence intervals (shaded areas) showing measured dependence between the size of DPPC liposomes and Young’s modulus estimated from Hertz model. Measured data along with the range of measured values are shown for liposomes filled with PBS and HA solution, denoted as DPPC and DPPC + HA, respectively. Indentation curves for selected liposomes A–D are shown in Figure 2.

## Data Availability

The original data presented in the study are openly available in Zenodo at https://zenodo.org/doi/10.5281/zenodo.11198627.

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
