# Peer review of "Size Matters: Rethinking Hertz Model Interpretation for Cell Mechanics Using AFM"

_ijms, 2024, doi:10.3390/ijms25137186_

Round 1

Reviewer 1 Report

Comments and Suggestions for Authors

The study titled "Size Matters: Rethinking Hertz Model Interpretation for Cell Mechanics Using AFM" presents findings on the Young’s modulus obtained from the Hertz model, demonstrating that is not only a property of the cell material, but also depends on the cell dimensions. That fact is obvious from the physics laws. Furthermore, using liposomes as cell models, with or without HA, doesn’t mimic the cell’s cytoskeleton.

It does not offer novel insights or advancements in the field and the calculus to obtain the young modulus are not specified. We are talking about very complex analysis without enough explanation. 

In more detail:

In line 18, "Originally developed to provide cell topographic images [1], it is now routinely applied to measure the mechanical properties of individual cells." is not true. 

Figure 2 have very bad data.

You should show all the equations and approximations used to the obtained values.

Comments on the Quality of English Language

It's easy to read. 

Author Response

Comment 1:

  • Reviewer: The study titled "Size Matters: Rethinking Hertz Model Interpretation for Cell Mechanics Using AFM" presents findings on the Young’s modulus obtained from the Hertz model, demonstrating that is not only a property of the cell material, but also depends on the cell dimensions. That fact is obvious from the physics laws.

  • Response: We acknowledge that cell size is a relevant factor in evaluating mechanical properties. However, contact mechanics theory suggests that for small deformations relative to cell size, the effect of dimensions can be neglected. Our study experimentally demonstrates that this assumption doesn't hold for AFM-based cell mechanics and quantifies the resulting error.

  • Text changes: page 2, lines 66-71: The assumption of cell size is stated explicitly in the revised manuscript. It is further supported by analysis of Hertz contact model in the Appendix A.2 (pages 8-9 , lines 287-318) where the Boussinesq's problem which addresses the stresses and deformations within an elastic half-space is explained.

Comment 2:

  • Reviewer: Furthermore, using liposomes as cell models, with or without HA, doesn’t mimic the cell’s cytoskeleton.

  • Your Response: Thank you for this suggestion. We have expanded the discussion section to emphasize the limitations of liposomes regarding the cytoskeleton. A new paragraph describes the dynamic role of the cytoskeleton, with references on active cytoskeletal influence and potential future models using reconstructed actin mesh (Lopes dos Santos et al., 2022; Rajagopal et al., 2018).

  • Text changes: page 5, lines 185-189 – a paragraph was added describing dynamic role of cytoskeleton. Two aditional references were added to point out the role of active cytoskeleton and to show possibility to upgrade the experimental model by reconstructing actin mesh within giant unilamellar vesicles.

  • Rajagopal, V.; Holmes, W.R.; Lee, P.V.S. Computational Modeling of Single-Cell Mechanics and Cytoskeletal Mechanobiology. WIREs Systems Biology and Medicine 2018, 10, e1407. https: //doi.org/10.1002/wsbm.1407.

  • Lopes dos Santos, R.; Campillo, C. Studying Actin-Induced Cell Shape Changes Using Giant Unilamellar Vesicles and Reconstituted Actin Networks. Biochemical Society Transactions 2022, 50, 1527–1539. https://doi.org/10.1042/BST20220900.

Comment 3:

  • Reviewer: It does not offer novel insights or advancements in the field and the calculus to obtain the young modulus are not specified. We are talking about very complex analysis without enough explanation. 

  • Your Response (Improved): We agree that the Hertz equation derivation is often missing in AFM literature. The complex derivation from contact mechanics textbooks (Hertz, 1882) can be challenging for a broad audience. Additionally, online forums show interest in the derivation (e.g., https://www.researchgate.net/post/How_to_derivate_the_equations_in_Hertzs_paper_1881_ON_THE_CONTACT_OF_ELASTIC_SOLIDS). To address this, we significantly expanded the paper by including a clear and complete derivation of the Hertz equation with relevant references (see Appendix).

  • Text changes: pages 6-10, new Appendix section was added that describes derivation of Hertz contact model. The appendix contains multiple references to relevant derivation of Hertz contact model.

Comment 4:

  • Reviewer: In line 18, "Originally developed to provide cell topographic images [1], it is now routinely applied to measure the mechanical properties of individual cells." is not true.

  • Your Response: You're right, the original statement was incorrect for general cell applications where tapping mode is used for soft cell surfaces. We've revised the definition of AFM development and included a reference for its original application in topographic imaging (Binnig et al., 1986).

  • Text changes: page 1, lines18-19: The text sounds in the revised manuscript: Originally developed to provide topographic images of solid surfaces, it is now routinely applied to measure the mechanical properties of individual cells.

    Reference to original application of AFM was added:

  • Binnig, G.; Quate, C.F.; Gerber, Ch. Atomic Force Microscope. Physical Review Letters 1986, 56, 930–933.

Comment 5:

  • Reviewer: Figure 2 has poor data quality.

  • Your Response: We've uploaded the figure as a vector image in the revised manuscript for improved resolution. The original data remains openly available at Zenodo (DOI: 10.5281/zenodo.11198627).

  • Text changes: Figure 2 is Figure 3 in the revised manuscript. Quality of the figure is improved.

Comment 6:

  • Reviewer: You should show all the equations and approximations used to the obtained values.

  • Your Response: We appreciate this suggestion. A new Appendix section has been added to the revised manuscript, containing the Hertz equation derivation, three new figures, and a clear statement of model limitations.

  • Text changes: pages 6-10, new Appendix section was added in the revised manuscript.

Reviewer 2 Report

Comments and Suggestions for Authors

Daniel and coworkers reports "Size Matters: Rethinking Hertz Model Interpretation for Cell Mechanics Using AFM" manuscript which studies the mechanical parts of cells. I recommend it for publication after lifting the following comments. Here are my comments;

1. The author would spent some time to revise the grammatical error and writing skills. The paragraphs or sentences are causing misunderstanding and not delivering the information in right way.

2. Please explain the Oliver and Pharr analysis in the introduction and provide a comparison with Hertz model. 

3. To attract reader's attention and ease understanding, I suggest the author can include one schematic presentation representing the overall idea of the manuscript.

Comments on the Quality of English Language

The manuscript contains a lot of grammatical errors and poor sentence/paragraph structure. It requires an extensive revision. 

Author Response

Comment 1:

  • Reviewer: The manuscript contains grammatical errors and writing issues that hinder clarity.

  • Response: Thank you for identifying this issue. We've carefully reviewed the manuscript and made significant edits to improve grammar, sentence structure, and overall clarity. The language in the manuscript was checked by English native speaker and edited for clarity.

Comment 2:

  • Reviewer: The Oliver-Pharr analysis is not explained and compared to the Hertz model.

  • Your Response: You're right, the Oliver-Pharr analysis was missing context. We've addressed this by:

    • Briefly explaining the principles of the Oliver-Pharr model in the Introduction section.

    • Including a relevant reference comparing the Hertz and Oliver-Pharr models.

    • Discussing the extension of Hertz contact model based on Oliver-Pharr analysis.

  • Text changes: Introduction page 2, lines 49-52, page 5, lines 172-178.

Comment 3:

  • Reviewer: A schematic or figure summarizing the main idea of the manuscript would improve readability.

  • Your Response: Thank you for this suggestion. We've incorporated a new figure (Figure 1 in the revised manuscript) that visually represents the overall concept and analysis of the study.

Round 2

Reviewer 1 Report

Comments and Suggestions for Authors

I have completed the review of your revised paper. I appreciate the efforts you have made in addressing the feedback provided.

Thank you for your attention to detail.